# Prior-free and prior-dependent regret bounds for Thompson Sampling

**Sébastien Bubeck, Che-Yu Liu**
Department of Operations Research and Financial Engineering,
Princeton University
sbubeck@princeton.edu, cheliu@princeton.edu

## Abstract

We consider the stochastic multi-armed bandit problem with a prior distribution on the reward distributions. We are interested in studying prior-free and prior-dependent regret bounds, very much in the same spirit than the usual distribution-free and distribution-dependent bounds for the non-Bayesian stochastic bandit. We first show that Thompson Sampling attains an optimal prior-free bound in the sense that for any prior distribution its Bayesian regret is bounded from above by $14\sqrt{nK}$. This result is unimprovable in the sense that there exists a prior distribution such that any algorithm has a Bayesian regret bounded from below by $\frac{1}{20}\sqrt{nK}$. We also study the case of priors for the setting of Bubeck et al. [2013] (where the optimal mean is known as well as a lower bound on the smallest gap) and we show that in this case the regret of Thompson Sampling is in fact uniformly bounded over time, thus showing that Thompson Sampling can greatly take advantage of the nice properties of these priors.

## 1 Introduction

In this paper we are interested in the Bayesian multi-armed bandit problem which can be described as follows. Let $\pi_0$ be a known distribution over some set $\Theta$, and let $\theta$ be a random variable distributed according to $\pi_0$. For $i \in [K]$, let $(X_{i,s})_{s\geq 1}$ be identically distributed random variables taking values in $[0, 1]$ and which are independent conditionally on $\theta$. Denote $\mu_i(\theta) := \mathbb{E}(X_{i,1}|\theta)$. Consider now an agent facing $K$ actions (or arms). At each time step $t = 1, \ldots n$, the agent pulls an arm $I_t \in [K]$. The agent receives the reward $X_{i,s}$ when he pulls arm $i$ for the $s^{th}$ time. The arm selection is based only on past observed rewards and potentially on an external source of randomness. More formally, let $(U_s)_{s\geq 1}$ be an i.i.d. sequence of random variables uniformly distributed on $[0, 1]$, and let $T_i(s) = \sum_{t=1}^s \mathbb{1}_{I_t=i}$, then $I_t$ is a random variable measurable with respect to $\sigma(I_1, X_{1,1}, \ldots, I_{t-1}, X_{I_{t-1}, T_{I_{t-1}}(t-1)}, U_t)$. We measure the performance of the agent through the Bayesian regret defined as

$$\text{BR}_n = \mathbb{E} \sum_{t=1}^n \left( \max_{i \in [K]} \mu_i(\theta) - \mu_{I_t}(\theta) \right),$$

where the expectation is taken with respect to the parameter $\theta$, the rewards $(X_{i,s})_{s\geq 1}$, and the external source of randomness $(U_s)_{s\geq 1}$. We will also be interested in the individual regret $R_n(\theta)$ which is defined similarly except that $\theta$ is fixed (instead of being integrated over $\pi_0$). When it is clear from the context we drop the dependency on $\theta$ in the various quantities defined above.

Given a prior $\pi_0$ the problem of finding an optimal strategy to minimize the Bayesian regret $\mathrm{BR}_n$ is a well defined optimization problem and as such it is merely a computational problem. On the other hand the point of view initially developed in Robbins [1952] leads to a learning problem. In this latter view the agent's strategy must have a low regret $R_n(\theta)$ for any $\theta \in \Theta$. Both formulations of the problem have a long history and we refer the interested reader to Bubeck and Cesa-Bianchi [2012] for a survey of the extensive recent literature on the learning setting. In the Bayesian setting a major breakthrough was achieved in Gittins [1979] where it was shown that when the prior distribution takes a *product form* an optimal strategy is given by the Gittins indices (which are relatively easy to compute). The product assumption on the prior means that the reward processes $(X_{i,s})_{s \geq 1}$ are independent across arms. In the present paper we are precisely interested in the situations where this assumption is not satisfied. Indeed we believe that one of the strength of the Bayesian setting is that one can incorporate prior knowledge on the arms in very transparent way. A prototypical example that we shall consider later on in this paper is when one knows the distributions of the arms up to a permutation, in which case the reward processes are strongly dependent.

In general without the product assumption on the prior it seems hopeless (from a computational perspective) to look for the optimal Bayesian strategy. Thus, despite being in a Bayesian setting, it makes sense to view it as a learning problem and to evaluate the agent's performance through its Bayesian regret. In this paper we are particularly interested in studying the Thompson Sampling strategy which was proposed in the very first paper on the multi-armed bandit problem Thompson [1933]. This strategy can be described very succinctly: let $\pi_t$ be the posterior distribution on $\theta$ given the history $H_t = (I_1, X_{1,1}, \ldots, I_{t-1}, X_{I_{t-1}, T_{I_{t-1}}(t-1)})$ of the algorithm up to the beginning of round $t$. Then Thompson Sampling first draws a parameter $\theta^{(t)}$ from $\pi_t$ (independently from the past given $\pi_t$) and it pulls $I_t \in \mathrm{argmax}_{i \in [K]} \mu_i(\theta^{(t)})$.

Recently there has been a surge of interest for this simple policy, mainly because of its flexibility to incorporate prior knowledge on the arms, see for example Chapelle and Li [2011]. For a long time the theoretical properties of Thompson Sampling remained elusive. The specific case of binary rewards with a Beta prior is now very well understood thanks to the papers Agrawal and Goyal [2012a], Kaufmann et al. [2012], Agrawal and Goyal [2012b]. However as we pointed out above here we are interested in proving regret bounds for the more realistic scenario where one runs Thompson Sampling with a hand-tuned prior distribution, possibly very different from a Beta prior. The first result in that spirit was obtained very recently by Russo and Roy [2013] who showed that for any prior distribution $\pi_0$ Thompson Sampling always satisfies $\mathrm{BR}_n \leq 5\sqrt{nK \log n}$. A similar bound was proved in Agrawal and Goyal [2012b] for the specific case of Beta prior[1]. Our first contribution is to show in Section 2 that the extraneous logarithmic factor in these bounds can be removed by using ideas reminiscent of the MOSS algorithm of Audibert and Bubeck [2009].

Our second contribution is to show that Thompson Sampling can take advantage of the properties of some non-trivial priors to attain much better regret guarantees. More precisely in Section 2 and 3 we consider the setting of Bubeck et al. [2013] (which we call the BPR setting) where $\mu^*$ and $\varepsilon > 0$ are known values such that for any $\theta \in \Theta$, first there is a unique best arm $\{i^*(\theta)\} = \mathrm{argmax}_{i \in [K]} \mu_i(\theta)$, and furthermore

$$\mu_{i^*(\theta)}(\theta) = \mu^*, \text{ and } \Delta_i(\theta) := \mu_{i^*(\theta)}(\theta) - \mu_i(\theta) \geq \varepsilon, \forall i \neq i^*(\theta).$$

In other words the value of the best arm is known as well as a non-trivial lower bound on the gap between the values of the best and second best arms. For this problem a new algorithm was proposed in Bubeck et al. [2013] (which we call the BPR policy), and it was shown that the BPR policy satisfies

$$R_n(\theta) = O\left(\sum_{i \neq i^*(\theta)} \frac{\log(\Delta_i(\theta)/\varepsilon)}{\Delta_i(\theta)} \log\log(1/\varepsilon)\right), \forall \theta \in \Theta, \forall n \geq 1.$$

Thus the BPR policy attains a regret uniformly bounded over time in the BPR setting, a feature that standard bandit algorithms such as UCB of Auer et al. [2002] cannot achieve. It is natural to view

the assumptions of the BPR setting as a prior over the reward distributions and to ask what regret guarantees attain Thompson Sampling in that situation. More precisely we consider Thompson Sampling with Gaussian reward distributions and uniform prior over the possible range of parameters. We then prove individual regret bounds for any sub-Gaussian distributions (similarly to Bubeck et al. [2013]). We obtain that Thompson Sampling uses optimally the prior information in the sense that it also attains uniformly bounded over time regret. Furthermore as an added bonus we remove the extraneous log-log factor of the BPR policy's regret bound.

The results presented in Section 2 and 3 can be viewed as a first step towards a better understanding of prior-dependent regret bounds for Thompson Sampling. Generalizing these results to arbitrary priors is a challenging open problem which is beyond the scope of our current techniques.

## 2 Optimal prior-free regret bound for Thompson Sampling

In this section we prove the following result.

**Theorem 1** *For any prior distribution $\pi_0$ over reward distributions in $[0,1]$, Thompson Sampling satisfies*

$$\mathrm{BR}_n \leq 14\sqrt{nK}.$$

Remark that the above result is unimprovable in the sense that there exist prior distributions $\pi_0$ such that for any algorithm one has $R_n \geq \frac{1}{20}\sqrt{nK}$ (see e.g. [Theorem 3.5, Bubeck and Cesa-Bianchi [2012]]). This theorem also implies an optimal rate of identification for the best arm, see Bubeck et al. [2009] for more details on this.

**Proof** We decompose the proof into three steps. We denote $i^*(\theta) \in \mathrm{argmax}_{i \in [K]} \mu_i(\theta)$, in particular one has $I_t = i^*(\theta_t)$.

**Step 1: rewriting of the Bayesian regret in terms of upper confidence bounds.** This step is given by [Proposition 1, Russo and Roy [2013]] which we reprove for sake of completeness. Let $B_{i,t}$ be a random variable measurable with respect to $\sigma(H_t)$. Note that by definition $\theta^{(t)}$ and $\theta$ are identically distributed conditionally on $H_t$. This implies by the tower rule:

$$\mathbb{E}B_{i^*(\theta),t} = \mathbb{E}B_{i^*(\theta^{(t)}),t} = \mathbb{E}B_{I_t,t}.$$

Thus we obtain:

$$\mathbb{E}\left(\mu_{i^*(\theta)}(\theta) - \mu_{I_t}(\theta)\right) = \mathbb{E}\left(\mu_{i^*(\theta)}(\theta) - B_{i^*(\theta),t}\right) + \mathbb{E}\left(B_{I_t,t} - \mu_{I_t}(\theta)\right).$$

Inspired by the MOSS strategy of Audibert and Bubeck [2009] we will now take

$$B_{i,t} = \widehat{\mu}_{i,T_i(t-1)} + \sqrt{\frac{\log_+\left(\frac{n}{KT_i(t-1)}\right)}{T_i(t-1)}},$$

where $\widehat{\mu}_{i,s} = \frac{1}{s}\sum_{t=1}^{s} X_{i,t}$, and $\log_+(x) = \log(x)\mathbb{1}_{x \geq 1}$. In the following we denote $\delta_0 = 2\sqrt{\frac{K}{n}}$. From now on we work conditionally on $\theta$ and thus we drop all the dependency on $\theta$.

**Step 2: control of $\mathbb{E}\left(\mu_{i^*(\theta)}(\theta) - B_{i^*(\theta),t}|\theta\right)$.** By a simple integration of the deviations one has

$$\mathbb{E}\left(\mu_{i^*} - B_{i^*,t}\right) \leq \delta_0 + \int_{\delta_0}^{1} \mathbb{P}(\mu_{i^*} - B_{i^*,t} \geq u)du.$$

Next we extract the following inequality from Audibert and Bubeck [2010] (see p2683–2684), for any $i \in [K]$,

$$\mathbb{P}(\mu_i - B_{i,t} \geq u) \leq \frac{4K}{nu^2}\log\left(\sqrt{\frac{n}{K}}u\right) + \frac{1}{nu^2/K - 1}.$$

Now an elementary integration gives

$$\int_{\delta_0}^{1} \frac{4K}{nu^2} \log\left(\sqrt{\frac{n}{K}}u\right) du = \left[-\frac{4K}{nu}\log\left(e\sqrt{\frac{n}{K}}u\right)\right]_{\delta_0}^{1} \leq \frac{4K}{n\delta_0}\log\left(e\sqrt{\frac{n}{K}}\delta_0\right) = 2(1+\log 2)\sqrt{\frac{K}{n}},$$

and

$$\int_{\delta_0}^{1} \frac{1}{nu^2/K - 1}du = \left[-\frac{1}{2}\sqrt{\frac{K}{n}}\log\left(\frac{\sqrt{\frac{n}{K}}u+1}{\sqrt{\frac{n}{K}}u-1}\right)\right]_{\delta_0}^{1} \leq \frac{1}{2}\sqrt{\frac{K}{n}}\log\left(\frac{\sqrt{\frac{n}{K}}\delta_0+1}{\sqrt{\frac{n}{K}}\delta_0-1}\right) = \frac{\log 3}{2}\sqrt{\frac{K}{n}}.$$

Thus we proved: $\mathbb{E}\left(\mu_{i^*(\theta)}(\theta) - B_{i^*(\theta),t}|\theta\right) \leq \left(2 + 2(1+\log 2) + \frac{\log 3}{2}\right)\sqrt{\frac{K}{n}} \leq 6\sqrt{\frac{K}{n}}.$

**Step 3: control of $\sum_{t=1}^{n} \mathbb{E}\left(B_{I_t,t} - \mu_{I_t}(\theta)|\theta\right)$.** We start again by integrating the deviations:

$$\mathbb{E}\sum_{t=1}^{n}(B_{I_t,t} - \mu_{I_t}) \leq \delta_0 n + \int_{\delta_0}^{+\infty}\sum_{t=1}^{n}\mathbb{P}(B_{I_t,t} - \mu_{I_t} \geq u)du.$$

Next we use the following simple inequality:

$$\sum_{t=1}^{n} \mathbb{1}\{B_{I_t,t} - \mu_{I_t} \geq u\} \leq \sum_{s=1}^{n}\sum_{i=1}^{K} \mathbb{1}\left\{\widehat{\mu}_{i,s} + \sqrt{\frac{\log_+\left(\frac{n}{Ks}\right)}{s}} - \mu_i \geq u\right\},$$

which implies

$$\sum_{t=1}^{n} \mathbb{P}(B_{I_t,t} - \mu_{I_t} \geq u) \leq \sum_{i=1}^{K}\sum_{s=1}^{n} \mathbb{P}\left(\widehat{\mu}_{i,s} + \sqrt{\frac{\log_+\left(\frac{n}{Ks}\right)}{s}} - \mu_i \geq u\right).$$

Now for $u \geq \delta_0$ let $s(u) = \lceil 3\log\left(\frac{nu^2}{K}\right)/u^2\rceil$ where $\lceil x\rceil$ is the smallest integer large than $x$. Let $c = 1 - \frac{1}{\sqrt{3}}$. It is is easy to see that one has:

$$\sum_{s=1}^{n} \mathbb{P}\left(\widehat{\mu}_{i,s} + \sqrt{\frac{\log_+\left(\frac{n}{Ks}\right)}{s}} - \mu_i \geq u\right) \leq \frac{3\log\left(\frac{nu^2}{K}\right)}{u^2} + \sum_{s=s(u)}^{n} \mathbb{P}\left(\widehat{\mu}_{i,s} - \mu_i \geq cu\right).$$

Using an integration already done in Step 2 we have

$$\int_{\delta_0}^{+\infty} \frac{3\log\left(\frac{nu^2}{K}\right)}{u^2} \leq 3(1+\log(2))\sqrt{\frac{n}{K}} \leq 5.1\sqrt{\frac{n}{K}}.$$

Next using Hoeffding's inequality and the fact that the rewards are in $[0,1]$ one has for $u \geq \delta_0$

$$\sum_{s=s(u)}^{n} \mathbb{P}\left(\widehat{\mu}_{i,s} - \mu_i \geq cu\right) \leq \sum_{s=s(u)}^{n} \exp(-2sc^2u^2)\mathbb{1}_{u\leq 1/c} \leq \frac{\exp(-12c^2\log 2)}{1 - \exp(-2c^2u^2)}\mathbb{1}_{u\leq 1/c}.$$

Now using that $1 - \exp(-x) \geq x - x^2/2$ for $x \geq 0$ one obtains

$$
\begin{aligned}
\int_{\delta_0}^{1/c} \frac{1}{1 - \exp(-2c^2u^2)}du &= \int_{\delta_0}^{1/(2c)} \frac{1}{1 - \exp(-2c^2u^2)}du + \int_{1/(2c)}^{1/c} \frac{1}{1 - \exp(-2c^2u^2)}du \\
&\leq \int_{\delta_0}^{1/(2c)} \frac{1}{2c^2u^2 - 2c^4u^4}du + \frac{1}{2c(1 - \exp(-1/2))} \\
&\leq \int_{\delta_0}^{1/(2c)} \frac{2}{3c^2u^2}du + \frac{1}{2c(1 - \exp(-1/2))} \\
&= \frac{2}{3c^2\delta_0} - \frac{4}{3c} + \frac{1}{2c(1 - \exp(-1/2))} \\
&\leq 1.9\sqrt{\frac{n}{K}}.
\end{aligned}
$$

Putting the pieces together we proved

$$\mathbb{E}\sum_{t=1}^{n}\left(B_{I_t,t} - \mu_{I_t}\right) \le 7.6\sqrt{nK},$$

which concludes the proof together with the results of Step 1 and Step 2. ∎

## 3 Thompson Sampling in the two-armed BPR setting

Following [Section 2, Bubeck et al. [2013]] we consider here the two-armed bandit problem with sub-Gaussian reward distributions (that is they satisfy $\mathbb{E}e^{\lambda(X-\mu)} \le e^{\lambda^2/2}$ for all $\lambda \in \mathbb{R}$) and such that one reward distribution has mean $\mu^*$ and the other one has mean $\mu^* - \Delta$ where $\mu^*$ and $\Delta$ are known values.

In order to derive the Thompson Sampling strategy for this problem we further assume that the reward distributions are in fact Gaussian with variance 1. In other words let $\Theta = \{\theta_1, \theta_2\}$, $\pi_0(\theta_1) = \pi_0(\theta_2) = 1/2$, and under $\theta_1$ one has $X_{1,s} \sim \mathcal{N}(\mu^*, 1)$ and $X_{2,s} \sim \mathcal{N}(\mu^* - \Delta, 1)$ while under $\theta_2$ one has $X_{2,s} \sim \mathcal{N}(\mu^*, 1)$ and $X_{1,s} \sim \mathcal{N}(\mu^* - \Delta, 1)$. Then a straightforward computation (using Bayes rule and induction) shows that one has for some normalizing constant $c > 0$:

$$\pi_t(\theta_1) = c\exp\left(-\frac{1}{2}\sum_{s=1}^{T_1(t-1)}(\mu^* - X_{1,s})^2 - \frac{1}{2}\sum_{s=1}^{T_2(t-1)}(\mu^* - \Delta - X_{2,s})^2\right),$$

$$\pi_t(\theta_2) = c\exp\left(-\frac{1}{2}\sum_{s=1}^{T_1(t-1)}(\mu^* - \Delta - X_{1,s})^2 - \frac{1}{2}\sum_{s=1}^{T_2(t-1)}(\mu^* - X_{2,s})^2\right).$$

Recall that Thompson Sampling draws $\theta^{(t)}$ from $\pi_t$ and then pulls the best arm for the environment $\theta^{(t)}$. Observe that under $\theta_1$ the best arm is arm 1 and under $\theta_2$ the best arm is arm 2. In other words Thompson Sampling draws $I_t$ at random with the probabilities given by the posterior $\pi_t$. This leads to a general algorithm for the two-armed BPR setting with sub-Gaussian reward distributions that we summarize in Figure 1. The next result shows that it attains optimal performances in this setting up to a numerical constant (see Bubeck et al. [2013] for lower bounds), for any sub-Gaussian reward distribution (not necessarily Gaussian) with largest mean $\mu^*$ and gap $\Delta$.

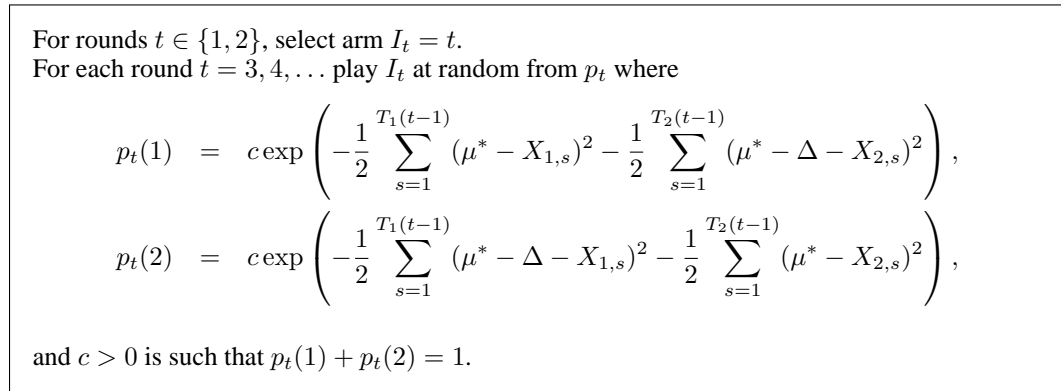

For rounds $t \in \{1, 2\}$, select arm $I_t = t$.
For each round $t = 3, 4, \ldots$ play $I_t$ at random from $p_t$ where

$$p_t(1) = c\exp\left(-\frac{1}{2}\sum_{s=1}^{T_1(t-1)}(\mu^* - X_{1,s})^2 - \frac{1}{2}\sum_{s=1}^{T_2(t-1)}(\mu^* - \Delta - X_{2,s})^2\right),$$

$$p_t(2) = c\exp\left(-\frac{1}{2}\sum_{s=1}^{T_1(t-1)}(\mu^* - \Delta - X_{1,s})^2 - \frac{1}{2}\sum_{s=1}^{T_2(t-1)}(\mu^* - X_{2,s})^2\right),$$

and $c > 0$ is such that $p_t(1) + p_t(2) = 1$.

**Figure 1:** Policy inspired by Thompson Sampling for the two-armed BPR setting.

**Theorem 2** *The policy of Figure 1 has regret bounded as $R_n \le \Delta + \frac{578}{\Delta}$, uniformly in $n$.*

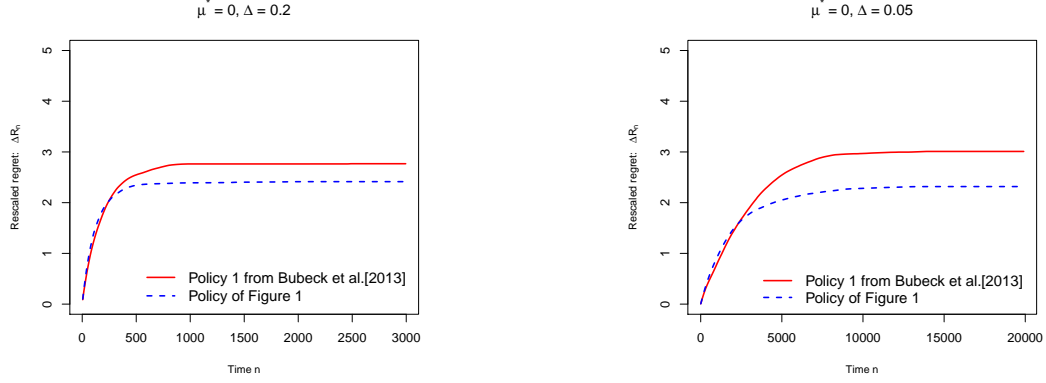

**Figure 2:** Empirical comparison of the policy of Figure 1 and Policy 1 of Bubeck et al. [2013] on Gaussian reward distributions with variance 1.

Note that we did not try to optimize the numerical constant in the above bound. Figure 2 shows an empirical comparison of the policy of Figure 1 with Policy 1 of Bubeck et al. [2013]. Note in particular that a regret bound of order $16/\Delta$ was proved for the latter algorithm and the (limited) numerical simulation presented here suggests that Thompson Sampling outperforms this strategy.

**Proof** Without loss of generality we assume that arm 1 is the optimal arm, that is $\mu_1 = \mu^*$ and $\mu_2 = \mu^* - \Delta$. Let $\widehat{\mu}_{i,s} = \frac{1}{s}\sum_{t=1}^{s} X_{i,t}$, $\widehat{\gamma}_{1,s} = \mu_1 - \widehat{\mu}_{1,s}$ and $\widehat{\gamma}_{2,s} = \widehat{\mu}_{2,s} - \mu_2$. Note that large (positive) values of $\widehat{\gamma}_{1,s}$ or $\widehat{\gamma}_{2,s}$ might mislead the algorithm into bad decisions, and we will need to control what happens in various regimes for these $\gamma$ coefficients. We decompose the proof into three steps.

**Step 1.** This first step will be useful in the rest of the analysis, it shows how the probability ratio of a bad pull over a good pull evolves as a function of the $\gamma$ coefficients introduced above. One has:

$$
\begin{aligned}
\frac{p_t(2)}{p_t(1)} &= \exp\left(-\frac{1}{2}\sum_{s=1}^{T_1(t-1)}\left[(\mu_2 - X_{1,s})^2 - (\mu_1 - X_{1,s})^2\right] - \frac{1}{2}\sum_{s=1}^{T_2(t-1)}\left[(\mu_1 - X_{2,s})^2 - (\mu_2 - X_{2,s})^2\right]\right) \\
&= \exp\left(-\frac{T_1(t-1)}{2}\left[\mu_2^2 - \mu_1^2 - 2(\mu_2 - \mu_1)\widehat{\mu}_{1,T_1(t-1)}\right] - \frac{T_2(t-1)}{2}\left[\mu_1^2 - \mu_2^2 - 2(\mu_1 - \mu_2)\widehat{\mu}_{2,T_2(t-1)}\right]\right) \\
&= \exp\left(-\frac{T_1(t-1)}{2}\left[\Delta^2 - 2\Delta(\mu_1 - \widehat{\mu}_{1,T_1(t-1)})\right] - \frac{T_2(t-1)}{2}\left[\Delta^2 - 2\Delta(\widehat{\mu}_{2,T_2(t-1)} - \mu_2)\right]\right) \\
&= \exp\left(-\frac{t\Delta^2}{2} + T_1(t-1)\Delta\widehat{\gamma}_{1,T_1(t-1)} + T_2(t-1)\Delta\widehat{\gamma}_{2,T_2(t-1)}\right).
\end{aligned}
$$

**Step 2.** We decompose the regret $R_n$ as follows:

$$
\begin{aligned}
\frac{R_n}{\Delta} &= 1 + \mathbb{E}\sum_{t=3}^{n}\mathbb{1}\{I_t = 2\} \\
&= 1 + \mathbb{E}\sum_{t=3}^{n}\mathbb{1}\left\{\widehat{\gamma}_{2,T_2(t-1)} > \frac{\Delta}{4}, I_t = 2\right\} + \mathbb{E}\sum_{t=3}^{n}\mathbb{1}\left\{\widehat{\gamma}_{2,T_2(t-1)} \le \frac{\Delta}{4}, \widehat{\gamma}_{1,T_1(t-1)} \le \frac{\Delta}{4}, I_t = 2\right\} \\
&\quad + \mathbb{E}\sum_{t=3}^{n}\mathbb{1}\left\{\widehat{\gamma}_{2,T_2(t-1)} \le \frac{\Delta}{4}, \widehat{\gamma}_{1,T_1(t-1)} > \frac{\Delta}{4}, I_t = 2\right\}.
\end{aligned}
$$

We use Hoeffding's inequality to control the first term:

$$
\mathbb{E}\sum_{t=3}^{n}\mathbb{1}\left\{\widehat{\gamma}_{2,T_2(t-1)} > \frac{\Delta}{4}, I_t = 2\right\} \le \mathbb{E}\sum_{s=1}^{n}\mathbb{1}\left\{\widehat{\gamma}_{2,s} > \frac{\Delta}{4}\right\} \le \sum_{s=1}^{n}\exp\left(-\frac{s\Delta^2}{32}\right) \le \frac{32}{\Delta^2}.
$$

For the second term, using the rewriting of Step 1 as an upper bound on $p_t(2)$, one obtains:

$$\mathbb{E}\sum_{t=3}^{n}\mathbb{1}\left\{\widehat{\gamma}_{2,T_2(t-1)}\leq\frac{\Delta}{4},\widehat{\gamma}_{1,T_1(t-1)}\leq\frac{\Delta}{4},I_t=2\right\}\;=\;\sum_{t=3}^{n}\mathbb{E}\left(p_t(2)\mathbb{1}\left\{\widehat{\gamma}_{2,T_2(t-1)}\leq\frac{\Delta}{4},\widehat{\gamma}_{1,T_1(t-1)}\leq\frac{\Delta}{4}\right\}\right)$$

$$\leq\;\sum_{t=3}^{n}\exp\left(-\frac{t\Delta^2}{4}\right)\leq\frac{4}{\Delta^2}.$$

The third term is more difficult to control, and we further decompose the corresponding event as follows:

$$\left\{\widehat{\gamma}_{2,T_2(t-1)}\leq\frac{\Delta}{4},\widehat{\gamma}_{1,T_1(t-1)}>\frac{\Delta}{4},I_t=2\right\}$$

$$\subset\left\{\widehat{\gamma}_{1,T_1(t-1)}>\frac{\Delta}{4},T_1(t-1)>t/4\right\}\cup\left\{\widehat{\gamma}_{2,T_2(t-1)}\leq\frac{\Delta}{4},I_t=2,T_1(t-1)\leq t/4\right\}.$$

The cumulative probability of the first event in the above decomposition is easy to control thanks to Hoeffding's maximal inequality[2] which states that for any $m\geq 1$ and $x>0$ one has

$$\mathbb{P}(\exists\,1\leq s\leq m\text{ s.t. }s\,\widehat{\gamma}_{1,s}\geq x)\leq\exp\left(-\frac{x^2}{2m}\right).$$

Indeed this implies

$$\mathbb{P}\left(\widehat{\gamma}_{1,T_1(t-1)}>\frac{\Delta}{4},T_1(t-1)>t/4\right)\leq\mathbb{P}\left(\exists\,1\leq s\leq t\text{ s.t. }s\,\widehat{\gamma}_{1,s}>\frac{\Delta t}{16}\right)\leq\exp\left(-\frac{t\Delta^2}{512}\right),$$

and thus

$$\mathbb{E}\sum_{t=3}^{n}\mathbb{1}\left\{\widehat{\gamma}_{1,T_1(t-1)}>\frac{\Delta}{4},T_1(t-1)>t/4\right\}\leq\frac{512}{\Delta^2}.$$

It only remains to control the term

$$\mathbb{E}\sum_{t=3}^{n}\mathbb{1}\left\{\widehat{\gamma}_{2,T_2(t-1)}\leq\frac{\Delta}{4},I_t=2,T_1(t-1)\leq t/4\right\}\;=\;\sum_{t=3}^{n}\mathbb{E}\left(p_t(2)\mathbb{1}\left\{\widehat{\gamma}_{2,T_2(t-1)}\leq\frac{\Delta}{4},T_1(t-1)\leq t/4\right\}\right)$$

$$\leq\;\sum_{t=3}^{n}\mathbb{E}\exp\left(-\frac{t\Delta^2}{4}+\Delta\max_{1\leq s\leq t/4}s\,\widehat{\gamma}_{1,s}\right),$$

where the last inequality follows from Step 1. The last step is devoted to bounding from above this last term.

**Step 3.** By integrating the deviations and using again Hoeffding's maximal inequality one obtains

$$\mathbb{E}\exp\left(\Delta\max_{1\leq s\leq t/4}s\,\widehat{\gamma}_{1,s}\right)\leq 1+\int_{1}^{+\infty}\mathbb{P}\left(\max_{1\leq s\leq\frac{t}{4}}s\,\widehat{\gamma}_{1,s}\geq\frac{\log x}{\Delta}\right)dx\leq 1+\int_{1}^{+\infty}\exp\left(-\frac{2(\log x)^2}{\Delta^2 t}\right)dx.$$

Now, straightforward computation gives

$$\sum_{t=3}^{n}\exp\left(-\frac{t\Delta^2}{4}\right)\left(1+\int_{1}^{+\infty}\exp\left(-\frac{2(\log x)^2}{\Delta^2 t}\right)dx\right)\;\leq\;\sum_{t=3}^{n}\exp\left(-\frac{t\Delta^2}{4}\right)\left(1+\sqrt{\frac{\pi\Delta^2 t}{2}}\exp\left(\frac{t\Delta^2}{8}\right)\right)$$

$$\leq\;\frac{4}{\Delta^2}+\int_{0}^{+\infty}\sqrt{\frac{\pi\Delta^2 t}{2}}\exp\left(-\frac{t\Delta^2}{8}\right)dt$$

$$\leq\;\frac{4}{\Delta^2}+\frac{16\sqrt{\pi}}{\Delta^2}\int_{0}^{+\infty}\sqrt{u}\exp(-u)\,du$$

$$\leq\;\frac{30}{\Delta^2}.$$

which concludes the proof by putting this together with the results of the previous step. $\blacksquare$

# 4 Optimal strategy for the BPR setting inspired by Thompson Sampling

In this section we consider the general BPR setting. That is the reward distributions are sub-Gaussian (they satisfy $\mathbb{E}e^{\lambda(X-\mu)} \leq e^{\lambda^2/2}$ for all $\lambda \in \mathbb{R}$), one reward distribution has mean $\mu^*$, and all the other means are smaller than $\mu^* - \varepsilon$ where $\mu^*$ and $\varepsilon$ are known values.

Similarly to the previous section we assume that the reward distributions are Gaussian with variance 1 for the derivation of the Thompson Sampling strategy (but we do not make this assumption for the analysis of the resulting algorithm). Then the set of possible parameters is described as follows:

$$\Theta = \cup_{i=1}^{K} \Theta_i \text{ where } \Theta_i = \{\theta \in \mathbb{R}^K \text{ s.t. } \theta_i = \mu^* \text{ and } \theta_j \leq \mu^* - \varepsilon \text{ for all } j \neq i\}.$$

Assuming a uniform prior over the index of the best arm, and a prior $\lambda$ over the mean of a suboptimal arm one obtains by Bayes rule that the probability density function of the posterior is given by:

$$d\pi_t(\theta) \propto \exp\left(-\frac{1}{2}\sum_{j=1}^{K}\sum_{s=1}^{T_j(t-1)}(X_{j,s}-\theta_j)^2\right) \prod_{j=1,j\neq i^*(\theta)}^{K} d\lambda(\theta_j).$$

Now remark that with Thompson Sampling arm $i$ is played at time $t$ if and only if $\theta^{(t)} \in \Theta_i$. In other words $I_t$ is played at random from probability $p_t$ where

$$p_t(i) = \pi_t(\Theta_i) \quad \propto \quad \exp\left(-\frac{1}{2}\sum_{s=1}^{T_i(t-1)}(X_{i,s}-\mu^*)^2\right) \prod_{j\neq i}\left[\int_{-\infty}^{\mu^*-\varepsilon}\exp\left(-\frac{1}{2}\sum_{s=1}^{T_j(t-1)}(X_{j,s}-v)^2\right)d\lambda(v)\right]$$

$$\propto \quad \frac{\exp\left(-\frac{1}{2}\sum_{s=1}^{T_i(t-1)}(X_{i,s}-\mu^*)^2\right)}{\int_{-\infty}^{\mu^*-\varepsilon}\exp\left(-\frac{1}{2}\sum_{s=1}^{T_i(t-1)}(X_{i,s}-v)^2\right)d\lambda(v)}.$$

Taking inspiration from the above calculation we consider the following policy, where $\lambda$ is the Lebesgue measure and we assume a slightly larger value for the variance (this is necessary for the proof).

---

For rounds $t \in [K]$, select arm $I_t = t$.
For each round $t = K+1, K+2, \ldots$ play $I_t$ at random from $p_t$ where

$$p_t(i) = c\frac{\exp\left(-\frac{1}{3}\sum_{s=1}^{T_i(t-1)}(X_{i,s}-\mu^*)^2\right)}{\int_{-\infty}^{\mu^*-\varepsilon}\exp\left(-\frac{1}{3}\sum_{s=1}^{T_i(t-1)}(X_{i,s}-v)^2\right)dv},$$

and $c > 0$ is such that $\sum_{i=1}^{K} p_t(i) = 1$.

---

**Figure 3:** Policy inspired by Thompson Sampling for the BPR setting.

The following theorem shows that this policy attains the best known performance for the BPR setting, shaving off a log-log term in the regret bound of the BPR policy.

**Theorem 3** *The policy of Figure 3 has regret bounded as* $R_n \leq \sum_{i:\Delta_i>0}\left(\Delta_i + \frac{80+\log(\Delta_i/\varepsilon)}{\Delta_i}\right)$, *uniformly in* $n$.

The proof of this result is fairly technical and it is deferred to the supplementary material.

## Footnotes

[1]Note however that the result of Agrawal and Goyal [2012b] applies to the individual regret $R_n(\theta)$ while the result of Russo and Roy [2013] only applies to the integrated Bayesian regret $\mathrm{BR}_n$.

[2]It is an easy exercise to verify that Azuma-Hoeffding holds for martingale differences with sub-Gaussian increments, which implies Hoeffding's maximal inequality for sub-Gaussian distributions.

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
