[Supplementary Material]

# Supplementary Material to 'Prior-free and prior-dependent regret bounds for Thompson Sampling'

**Sébastien Bubeck, Che-Yu Liu**
Department of Operations Research and Financial Engineering,
Princeton University
sbubeck@princeton.edu, cheliu@princeton.edu

## Proof of Theorem 3

The general structure of the proof is superficially similar to the proof of Theorem 2 but many details are different. Without loss of generality we assume that arm 1 is the optimal arm, that is $\mu_1 = \mu^*$ and $\forall i \geq 2, \mu_i = \mu^* - \Delta_i$. Let $\widehat{\gamma}_{1,s} = \mu_1 - \widehat{\mu}_{1,s}$ and $\widehat{\gamma}_{i,s} = \widehat{\mu}_{i,s} - \mu_i$ for $i \geq 2$. We decompose the proof into four steps.

**Step 1: Rewriting of the ratio $\frac{p_{i,t}}{p_{1,t}}$.** Let $i \geq 2$, the following rewriting will be useful in the rest of the proof:

$$
\begin{aligned}
\frac{p_t(i)}{p_t(1)} &= \frac{\int_{-\infty}^{\mu_1-\varepsilon} \exp\left(-\frac{1}{3}\sum_{s=1}^{T_1(t-1)}(X_{1,s}-v)^2 - (X_{1,s}-\mu_1)^2\right)\, dv}{\int_{-\infty}^{\mu_1-\varepsilon} \exp\left(-\frac{1}{3}\sum_{s=1}^{T_i(t-1)}(X_{i,s}-v)^2 - (X_{i,s}-\mu_1)^2\right)\, dv} \\
&= \frac{\int_{-\infty}^{\mu_1-\varepsilon} \exp\left(-\frac{T_1(t-1)}{3}(\widehat{\mu}_{1,T_1(t-1)}-v)^2 - (\widehat{\mu}_{1,T_1(t-1)}-\mu_1)^2\right)\, dv}{\int_{-\infty}^{\mu_1-\varepsilon} \exp\left(-\frac{T_i(t-1)}{3}(\widehat{\mu}_{i,T_i(t-1)}-v)^2 - (\widehat{\mu}_{i,T_i(t-1)}-\mu_1)^2\right)\, dv} \\
&= \frac{\int_{-\widehat{\gamma}_{1,T_1(t-1)}+\varepsilon}^{+\infty} \exp\left(-\frac{T_1(t-1)}{3}(v^2 - \widehat{\gamma}^2_{1,T_1(t-1)})\right)\, dv}{\int_{\widehat{\gamma}_{i,T_i(t-1)}-\Delta_i+\varepsilon}^{+\infty} \exp\left(-\frac{T_i(t-1)}{3}(v^2 - (\widehat{\gamma}_{i,T_i(t-1)}-\Delta_i)^2)\right)\, dv},
\end{aligned}
$$

where the last step follows by a simple change of variable.

**Step 2: Decomposition of $R_n$.** For $i \geq 2$. Let $A_i = \lceil \frac{6}{\Delta_i^2} \log(\frac{e^6 \Delta_i}{\varepsilon}) \rceil$ where $\lceil x \rceil$ is the smallest integer larger than $x$. We decompose the regret $R_n$ as follows.

$$
\begin{aligned}
R_n &= \sum_{i=2}^{K} \left( \Delta_i + \Delta_i \mathbb{E} \sum_{t=K+1}^{n} \mathbb{1}\{I_t = i\} \right) \\
&\leq \sum_{i=2}^{K} \Delta_i \left( A_i + \mathbb{E} \sum_{t=K+1}^{n} \mathbb{1}\{T_i(t-1) \geq A_i, I_t = i\} \right) \\
&= \sum_{i=2}^{K} \Delta_i \left( A_i + \mathbb{E} \sum_{t=K+1}^{n} \mathbb{1}\left\{ \widehat{\gamma}_{i,T_i(t-1)} > \frac{\Delta_i}{4}, T_i(t-1) \geq A_i, I_t = i \right\} \right. \\
&\qquad \left. + \mathbb{E} \sum_{t=K+1}^{n} \mathbb{1}\left\{ \widehat{\gamma}_{i,T_i(t-1)} \leq \frac{\Delta_i}{4}, T_i(t-1) \geq A_i, I_t = i \right\} \right).
\end{aligned}
$$

The first expectation can be bounded by using Hoeffding's inequality.

$$
\mathbb{E} \sum_{t=K+1}^{n} \mathbb{1}\left\{ \widehat{\gamma}_{i,T_i(t-1)} > \frac{\Delta_i}{4}, T_i(t-1) \geq A_i, I_t = i \right\} \leq \mathbb{E} \sum_{s=1}^{n} \mathbb{1}\left\{ \widehat{\gamma}_{i,s} > \frac{\Delta_i}{4} \right\} \leq \sum_{s=1}^{n} \exp\left( -\frac{s\Delta_i^2}{32} \right) \leq \frac{32}{\Delta_i^2}.
$$

The second expectation is more difficult to bound from above and the next two steps are dedicated to this task.

**Step 3: Analysis of $\sum_{t=K+1}^{n} \mathbb{E} \mathbb{1}\left\{ \widehat{\gamma}_{i,T_i(t-1)} \leq \frac{\Delta_i}{4}, T_i(t-1) \geq A_i, I_t = i \right\}$.** Clearly by definition of the policy one has

$$
\begin{aligned}
&\sum_{t=K+1}^{n} \mathbb{E} \mathbb{1}\left\{ \widehat{\gamma}_{i,T_i(t-1)} \leq \frac{\Delta_i}{4}, T_i(t-1) \geq A_i, I_t = i \right\} \\
&= \sum_{t=K+1}^{n} \mathbb{E} \left[ \frac{p_t(i)}{p_t(1)} \mathbb{1}\left\{ \widehat{\gamma}_{i,T_i(t-1)} \leq \frac{\Delta_i}{4}, T_i(t-1) \geq A_i, I_t = 1 \right\} \right].
\end{aligned}
$$

We have now to control the term $\frac{p_t(i)}{p_t(1)}$ on the event $\{\widehat{\gamma}_{i,T_i(t-1)} \leq \frac{\Delta_i}{4}, T_i(t-1) \geq A_i\}$. The following bounds on the tail of the standard Gaussian distribution will be useful, for any $x > 0$ one has

$$
\frac{1}{x} e^{-\frac{1}{2}x^2} \geq \int_x^{+\infty} e^{-\frac{1}{2}v^2}\, dv \geq \frac{1}{x}\left(1 - \frac{1}{x^2}\right) e^{-\frac{1}{2}x^2}.
$$

Now one has

$$
\begin{aligned}
&\int_{\widehat{\gamma}_{i,T_i(t-1)}-\Delta_i+\varepsilon}^{+\infty} e^{-\frac{1}{3}T_i(t-1)(v^2 - (\widehat{\gamma}_{i,T_i(t-1)} - \Delta_i)^2)}\, dv \\
&= e^{\frac{1}{3}T_i(t-1)(\widehat{\gamma}_{i,T_i(t-1)} - \Delta_i)^2} \int_{\widehat{\gamma}_{i,T_i(t-1)}-\Delta_i+\varepsilon}^{+\infty} e^{-\frac{1}{3}T_i(t-1)v^2}\, dv \\
&\geq e^{\frac{3}{16}T_i(t-1)\Delta_i^2} \int_{\frac{\Delta_i}{4}}^{+\infty} e^{-\frac{1}{3}T_i(t-1)v^2}\, dv \\
&= e^{\frac{3}{16}T_i(t-1)\Delta_i^2} \cdot \sqrt{\frac{3}{2T_i(t-1)}} \cdot \int_{\frac{\Delta_i}{4}\sqrt{\frac{2T_i(t-1)}{3}}}^{+\infty} e^{-\frac{1}{2}v^2}\, dv \\
&\geq e^{\frac{3}{16}T_i(t-1)\Delta_i^2} \cdot \frac{6}{\Delta_i T_i(t-1)}\left(1 - \frac{24}{\Delta_i^2 T_i(t-1)}\right) e^{-\frac{1}{48}T_i(t-1)\Delta_i^2} \\
&\geq e^{\frac{1}{6}T_i(t-1)\Delta_i^2} \cdot \frac{2}{\Delta_i T_i(t-1)},
\end{aligned}
$$

where the last step follows from

$$T_i(t-1) \geq A_i \geq \frac{6}{\Delta_i^2} \log\left(\frac{e^6 \Delta_i}{\varepsilon}\right) \geq \frac{36}{\Delta_i^2}.$$

Next, using the fact that the function $x \to \frac{1}{x} e^{\frac{1}{6}x\Delta_i^2}$ is increasing on $[\frac{6}{\Delta_i^2}, +\infty)$, we get

$$
\begin{aligned}
\left( \int_{\widehat{\gamma}_{i,T_i(t-1)} - \Delta_i + \varepsilon}^{+\infty} e^{-\frac{1}{3}T_i(t-1)(v^2 - (\widehat{\gamma}_{i,T_i(t-1)} - \Delta)^2)} \, dv \right)^{-1} &\leq \left( e^{\frac{1}{6}T_i(t-1)\Delta^2} \cdot \frac{2}{\Delta_i T_i(t-1)} \right)^{-1} \\
&\leq e^{-\frac{1}{6}\Delta_i^2 \left( \frac{6}{\Delta_i^2} \log\left(\frac{e^6\Delta_i}{\varepsilon}\right) \right)} \frac{\Delta_i}{2} \frac{6}{\Delta_i^2} \log\left(\frac{e^6\Delta}{\varepsilon}\right) \\
&= \frac{3}{e^6} \frac{\varepsilon}{\Delta_i^2} \log\left(\frac{e^6\Delta_i}{\varepsilon}\right).
\end{aligned}
$$

Plugging into the expression of $\frac{p_t(i)}{p_t(1)}$, we obtain

$$
\begin{aligned}
&\sum_{t=K+1}^{n} \mathbb{E}\, \mathbb{1}_{\{\widehat{\gamma}_{i,T_i(t-1)} \leq \frac{\Delta_i}{4}, T_i(t-1) \geq A_i, I_t = i\}} \\
&= \sum_{t=K+1}^{n} \mathbb{E}\left[ \frac{p_{i,t}}{p_{1,t}} \mathbb{1}_{\{\widehat{\gamma}_{i,T_i(t-1)} \leq \frac{\Delta_i}{4}, T_i(t-1) \geq A_i, I_t = 1\}} \right] \\
&\leq \left( \sum_{t=K+1}^{n} \mathbb{E}\left[ \int_{-\widehat{\gamma}_{1,T_1(t-1)} + \varepsilon}^{+\infty} e^{-\frac{1}{3}T_1(t-1)(v^2 - \widehat{\gamma}_{1,T_1(t-1)}^2)} \, dv \mathbb{1}_{\{I_t = 1\}} \right] \right) \frac{3}{e^6} \frac{\varepsilon}{\Delta_i^2} \log\left(\frac{e^6\Delta_i}{\varepsilon}\right). \\
&\leq \left( \sum_{t=1}^{+\infty} \mathbb{E}\left[ \int_{-\widehat{\gamma}_{1,t} + \varepsilon}^{+\infty} e^{-\frac{1}{3}t(v^2 - \widehat{\gamma}_{1,t}^2)} \, dv \right] \right) \frac{3}{e^6} \frac{\varepsilon}{\Delta_i^2} \log\left(\frac{e^6\Delta_i}{\varepsilon}\right).
\end{aligned}
$$

**Step 4: Control of** $\sum_{t=1}^{+\infty} \mathbb{E}\left[ \int_{-\widehat{\gamma}_{1,t}+\varepsilon}^{+\infty} e^{-\frac{1}{3}t(v^2 - \widehat{\gamma}_{1,t}^2)} \, dv \right]$.
First, observe that

$$
\begin{aligned}
&\sum_{t=1}^{+\infty} \mathbb{E}\left[ \int_{-\widehat{\gamma}_{1,t}+\varepsilon}^{+\infty} e^{-\frac{1}{3}t(v^2 - \widehat{\gamma}_{1,t}^2)} \, dv \right] \\
&\leq \sum_{t=1}^{+\infty} \mathbb{E}\left[ \int_{-\widehat{\gamma}_{1,t}+\varepsilon}^{+\infty} e^{-\frac{1}{3}tv^2} \, dv \cdot e^{\frac{1}{3}t\widehat{\gamma}_{1,t}^2} \mathbb{1}_{\{|\widehat{\gamma}_{1,t}| \leq \frac{\varepsilon}{3}\}} \right] + \sum_{t=1}^{+\infty} \mathbb{E}\left[ \int_{-\widehat{\gamma}_{1,t}+\varepsilon}^{+\infty} e^{-\frac{1}{3}tv^2} \, dv \cdot e^{\frac{1}{3}t\widehat{\gamma}_{1,t}^2} \mathbb{1}_{\{|\widehat{\gamma}_{1,t}| \geq \frac{\varepsilon}{3}\}} \right] \\
&\leq \sum_{t=1}^{+\infty} \int_{\frac{2}{3}\varepsilon}^{+\infty} e^{-\frac{1}{3}tv^2} \, dv \cdot e^{\frac{1}{27}t\varepsilon^2} + \sum_{t=1}^{+\infty} \int_{-\infty}^{+\infty} e^{-\frac{1}{3}tv^2} \, dv \cdot \mathbb{E}\left[ e^{\frac{1}{3}t\widehat{\gamma}_{1,t}^2} \mathbb{1}_{\{|\widehat{\gamma}_{1,t}| \geq \frac{\varepsilon}{3}\}} \right].
\end{aligned}
$$

The first term is straightforward to compute:

$$
\begin{aligned}
\sum_{t=1}^{+\infty} \int_{\frac{2}{3}\varepsilon}^{+\infty} e^{-\frac{1}{3}tv^2} \, dv \cdot e^{\frac{1}{27}t\varepsilon^2} &= \int_{\frac{2}{3}\varepsilon}^{+\infty} \sum_{t=1}^{+\infty} e^{-\frac{1}{3}t(v^2 - \frac{1}{9}\varepsilon^2)} \, dv \\
&\leq \int_{\frac{2}{3}\varepsilon}^{+\infty} \frac{3}{v^2 - \frac{1}{9}\varepsilon^2} \, dv \leq \frac{9\log 3}{2\varepsilon}
\end{aligned}
$$

For the second term, we first integrate the deviations and we use Hoeffding's inequality to obtain

$$
\begin{aligned}
\mathbb{E}\left[e^{\frac{1}{3}t\widehat{\gamma}_{1,t}^2}\mathbb{1}_{\{|\widehat{\gamma}_{1,t}|\geq\frac{\varepsilon}{3}\}}\right] &\leq e^{\frac{1}{3}t(\frac{\varepsilon}{3})^2}\mathbb{P}(|\widehat{\gamma}_{1,t}|\geq\frac{\varepsilon}{3})+\int_{e^{\frac{1}{3}t(\frac{\varepsilon}{3})^2}}^{+\infty}\mathbb{P}(e^{\frac{1}{3}t\widehat{\gamma}_{1,t}^2}\geq x)\,dx\\
&\leq 2e^{-\frac{1}{54}t\varepsilon^2}+\int_{e^{\frac{1}{27}t\varepsilon^2}}^{+\infty}\mathbb{P}\left(|\widehat{\gamma}_{1,t}|\geq\sqrt{\frac{3\log x}{t}}\right)\,dx\\
&\leq 2e^{-\frac{1}{54}t\varepsilon^2}+2\int_{e^{\frac{1}{27}t\varepsilon^2}}^{+\infty}e^{-\frac{3}{2}\log x}\,dx\\
&\leq 6e^{-\frac{1}{54}t\varepsilon^2},
\end{aligned}
$$

which yields

$$
\begin{aligned}
\sum_{t=1}^{+\infty}\int_{-\infty}^{+\infty}e^{-\frac{1}{3}tv^2}\,dv\cdot\mathbb{E}\left[e^{\frac{1}{3}t\widehat{\gamma}_{1,t}^2}\mathbb{1}_{\{|\widehat{\gamma}_{1,t}|\geq\frac{\varepsilon}{3}\}}\right] &\leq \int_{-\infty}^{+\infty}6\sum_{t=1}^{+\infty}e^{-\frac{1}{3}t(v^2+\frac{1}{18}\varepsilon^2)}\,dv\\
&\leq \int_{-\infty}^{+\infty}18\frac{1}{v^2+\frac{1}{18}\varepsilon^2}\,dv=\frac{54\sqrt{2}\pi}{\varepsilon}.
\end{aligned}
$$

Putting together all the steps finishes the proof.