[Reviews · NeurIPS 2013]

Submitted by Assigned_Reviewer_6

This paper makes two contributions to the analysis of Thompson Sampling. First, it removes a logarithmic factor from existing bounds on the algorithm's Bayesian regret. These bounds hold under an arbitrary prior distribution. Second, the paper considers a problem where the mean reward from the optimal arm is known, as is that of any suboptimal arm, but the identity of the optimal arm is unknown. In this setting, it was shown that if a particular prior is used then the regret of Thompson sampling is bounded uniformly over time.

The paper is written clearly, and overall, was very well executed. Each of the main bounds in the paper is order-optimal. The author's first bound applies to a very broad class of problems, and is therefore valuable, even if it only removes an extraneous logarithmic factor from prior results. The BPR problem setting is much more specialized, and it's less clear how to motivate such a problem formulation. Still, it is interesting that one does not need to design a new algorithm to address this problem, and that Thompson Sampling itself can exploit its special structure.

While each result is solid, neither constitutes a truly novel and significant contribution to the Thompson sampling literature. Further, the two contributions are quite disconnected, and it's not clear they belong in the same paper. Still, I think the paper does enough to merit acceptance.
Summary: This paper makes two decent contributions to the literature, and deserves to be accepted.

Submitted by Assigned_Reviewer_7

The paper proves new bounds on the Bayesian regret of the Thompson sampling algorithm. Based on the idea of Thompson sampling it also introduces a new algorithm for the case that the average reward of the optimal arm as well as some lower bound on the minimum gap is known. It shows that, in this case, it is possible to achieve a bounded regret which scales slightly better (in terms of \epsilon) than the best previous result (and matches the lower bound).


The paper and the technical proofs are very well organized and easy to follow. I checked the proofs of both theorems and I could not find any flaw in them. Also the results are fairly interesting, especially the observation that it is possible to achieve a bounded regret in the case that we know the average reward and the minimum gap can be very useful in real applications. I was wondering whether this bound can also be applied to the case that the prior knowledge is in the form of some interval over \mu_*? Also I wonder whether it is possible to achieve a similar Bayesian regret as the one in this paper for other bandit algorithms?: we know that a modified version of UCB algorithm can achieve a regret of O(\sqrt{nK}) w.h.p. I think it shouldn't be difficult to transform that bound to the Bayesian regret of a same order since typically high probability bounds provide stronger performance guarantees than the expected regret bounds. Of course incorporating the prior knowledge in UCB is not trivial so maybe such a result would not be very interesting from the practical point of view but as a theoretical practice it would be still interesting to see whether other algorithms can achieve this minimax regret or not.

Overall, I think the paper includes some interesting new results and the new algorithm may be useful for problems which we have some prior over the optimal arm, though the improvement w.r.t. the state-of-the-art is incremental.

Minor:

Line 172: is is-> is
Summary: The paper includes some interesting new results, though the improvement w.r.t. the state-of-the-art is incremental.

Submitted by Assigned_Reviewer_8

Paper summary:
--------------
The paper provides an analysis Bayesian regret of Thompson sampling (where Bayesian regret is defined as an average regret weighted according to a prior distribution over the parameters) and regret bounds for Thompson sampling in the setting of the paper of Bubeck, Perchet, and Rigollet 2013, where it is assumed that the mean reward of the best arm and the gap to the second-best arm are known, but the identity of the best arm is unknown. In this setting the authors improve the upper bound by a log(log(1/epsilon)) factor, thus closing the gap with the lower bound down to a constant.

Quality, clarity, originality, and significance:
------------------------------------------------
The paper is very technical and impractical. Most of the paper is devoted to technical proofs that could be moved into appendices, whereas no motivation whatsoever is provided for the model studied.

Main criticism:
---------------
As already said, my main criticism is about lack of motivation behind studying the model studied by the authors. In the first setting considered the authors assume that a prior over parameters (arm biases) is provided and study the performance of the model with respect to this prior. It would be good to have some examples, where this assumption and evaluation measure are plausible. In the second and third settings the authors go into further extreme and assume that the bias of the best arm and the gap to the second-best arm are known and only the identity of the best arm is unknown. Once again, there are no examples, where this assumption is satisfied.

There are also no examples or discussion of how the results of the paper could be useful for solving some other theoretical problems. So we have neither practical nor theoretical motivation. From looking at the details of the proofs (which are the main part of the paper) it is hard to see whether any of the ideas could be reused in some other settings.

========================
Response to the rebuttal
========================
I thank the authors for their response. I would like to point out that providing the reader a motivation why to read your paper is not called "waste of time and space". Going straight to the formulas could probably be considered marginally acceptable at COLT (although I would criticize it there as well), but at NIPS you have a much broader audience and not everybody is dealing with your specific problem on a daily basis. I would like to point out that none of my questions were actually addressed. I asked to give at least one specific example (out of "hundreds of papers written since the 70s"), where evaluation of an algorithm with respect to a known prior over the parameters would be a reasonable thing to do, but got none. I asked to give some motivation for your study of the BPR setting - you are referring me for an example to the review of Reviewer_7, but Reviewer_7 just says that it "can be useful in real applications" and provides no example, so I cannot consider it as a satisfactory answer to my question. I have read the BPR paper and it provides very clear motivation for studying the model in the setting they study it; they have very clear proofs with ideas that are easy to reuse; and they have a lot of results that go far beyond the BPR model, such as lower bounds. But even after reading the BPR paper I see no motivation to study the problem in the setting you study it and it is your job to explain why is it interesting. And it is not explained on page 2 you are referring me to.

And one more point - you did not discuss how your analysis of Thompson sampling in the Bayesian setting relates to Bayes-UCB.
Summary: The paper is very technical and lacks connection to practical or other theoretical problems. There is no sufficient explanation of why the results presented in the paper are interesting.
Author Feedback

Author rebuttal: Reviewer_6:
Thank you for your kind words! We believe that our two results form a cohesive paper in the sense of the objective described in line 56-59. Indeed the trend recently has been to prove that TS works well for a sort of 'agnostic' prior (the Beta prior). On the contrary we believe that the strength of the Bayesian formulation lies in its flexibility to incorporate complex prior knowledge in a very transparent way. With that point of view the results on TS with agnostic prior are not interesting. Both of our theorems go in the direction of showing that TS works in fact with any prior, and that for some of them it can even exploit optimally the specific properties of the prior. We illustrate this with a prior for the Bubeck, Perchet, Rigollet setting. We totally agree with you that this is only a first step, but we hope that a NIPS paper on this topic could greatly stimulate the community to make further progress on this important problem, perhaps eventually leading to a characterization of the priors for which TS is optimal. Note also that while the BPR setting is indeed very specialized Reviewer_7 points out that it "[...] can be very useful in real applications" and we completely agree with him.

Reviewer_7:
Thank you for your review! Since you have technical questions we directly answer them.
- It is possible to deal with only an interval for \mu^* rather than an exact value but this would add an extra layer of notation so we decided to restrict to the simplest framework.
- It is also possible to obtain high probability statements through a martingale concentration argument but again for sake of simplicity (since NIPS papers are short) we have decided to not include this result.
Finally let us make one minor remark: we view the improvements in the bounds as an interesting observation but to us the real contribution of the paper lies in the fact that we show for the first time the potential power of adaptivity of TS for specific priors, see the reply to Reviewer_6 for more on this.

Reviewer_8:
The Bayesian framework for the multi-armed bandit has been actively studied (with hundreds of papers) from the 70s to the 90s, this is why we decided to not waste time and space with examples where this framework is of utter practical importance (but there are plentiful). On the other hand it is true that the BPR setting is much more recent, but Reviewer_7 points out that it "[...] can be very useful in real applications". Finally we do not understand why you say that we have no theoretical/practical motivation while we provide both of them on page 2. In a nutshell we are interested in showing that TS can adapt optimally to potentially complex and non-product priors. This fact has great practical implications as one of the main reason for the recent surge of interest in TS is that it allows to incorporate prior knowledge in a transparent way (which is not the case with UCB for instance).